# Evolving Concepts on Inflammatory Biomarkers and Malnutrition in Chronic Kidney Disease

**DOI:** 10.3390/nu14204297

**Published:** 2022-10-14

**Authors:** Fredzzia Graterol Torres, María Molina, Jordi Soler-Majoral, Gregorio Romero-González, Néstor Rodríguez Chitiva, Maribel Troya-Saborido, Guillem Socias Rullan, Elena Burgos, Javier Paúl Martínez, Marina Urrutia Jou, Carles Cañameras, Josep Riera Sadurní, Anna Vila, Jordi Bover

**Affiliations:** 1Nephrology Department, University Hospital Germans Trias i Pujol (HGTiP) & REMAR-IGTP Group, Germans Trias i Pujol Research Institute (IGTP), Can Ruti Campus, 08916 Badalona, Spain; 2Medicine Department, Universitat Autònoma de Barcelona, 08193 Barcelona, Spain; 3International Renal Research Institute of Vicenza, 36100 Vicenza, Italy; 4Endocrinology and Nutrition Department, Hospital Universitari Germans Trias i Pujol, 08916 Badalona, Spain

**Keywords:** chronic kidney disease, CKD, malnutrition, sarcopenia, cytokines, inflammation, proinflammatory cytokines, protein-energy wasting, malnutrition, malnutrition inflammation cachexia syndrome, MICS, malnutrition-inflammation score, MIS

## Abstract

While patient care, kidney replacement therapy, and transplantation techniques for chronic kidney disease (CKD) have continued to progress, the incidence of malnutrition disorders in CKD appears to have remained unchanged over time. However, there is now a better understanding of the underlying pathophysiology according to the disease background, disease stage, and the treatment received. In CKD patients, the increased production of proinflammatory cytokines and oxidative stress lead to a proinflammatory milieu that is at least partially responsible for the increased morbidity and mortality in this patient population. New insights into the pathogenic role of innate immunity and the proinflammatory cytokine profile, characterized, for instance, by higher levels of IL-6 and TNF-α, explain some of the clinical and laboratory abnormalities observed in these patients. In this article, we will explore currently available nutritional–inflammatory biomarkers in distinct CKD populations (hemodialysis, peritoneal dialysis, transplantation) with a view to evaluating their efficacy as predictors of malnutrition and their involvement in the common proinflammatory process. Although there is a direct relationship between inflammatory-nutritional status, signs and symptoms [e.g., protein-energy wasting (PEW), anorexia], and comorbidities (e.g., atheromatosis, atherosclerosis), we are in need of clearly standardized markers for nutritional-inflammatory assessment to improve their performance and design appropriate bidirectional interventions.

## 1. Introduction

Especially in the final stages of chronic kidney disease (CKD), poorly controlled patients frequently present a worsening of their nutritional status [1], as reflected by the depletion of body proteins and energy reserves. This population has muscle wasting, sarcopenia, and cachexia, contributing to their frailty [2], and the presence of malnutrition is one of the stronger predictors of morbidity and mortality in end-stage kidney disease (ESKD) patients [3]. A state of disordered catabolism that is induced by uraemia and is characterized by a loss of body fat, visceral fat, and muscle constitutes a condition known as protein-energy wasting (PEW) [4,5]. Retrospective epidemiological studies have strongly linked PEW with mortality in CKD and ESKD [5]. There are currently no standardized markers for nutritional-inflammatory bidirectional assessment in this population, and it is not clear which measurements offer predictably superior performance. In fact, there is a complex interplay between overlapping nutritional and inflammatory parameters during CKD, and there is a need for further assessment and interpretation of these parameters as they relate to identifying and preventing malnutrition in particular. Therefore, this review aims to explore the currently available nutritional–inflammatory biomarkers in adults with a view to evaluating their efficacy as predictors of malnutrition and their involvement in the proinflammatory process in this patient population.

## 2. Mechanisms of Inflammation in CKD and Their Relation to Malnutrition

Many factors contribute to the chronic inflammatory state in CKD. The increased production of proinflammatory cytokines and oxidative stress [6,7,8] mediated by the activation of macrophages and monocytes constitute the fundamental basis for the initiation of the inflammatory cascade. As already mentioned, inflammation is associated with increased morbidity and mortality in patients with CKD [9], and persistent inflammation is associated with adverse cardiovascular outcomes (e.g., atheromatosis and atherosclerosis) and a poor prognosis [2,7,8]. Related to malnutrition/PEW, also referred to as malnutrition–inflammation–cachexia syndrome (MICS). In addition to the diagnostic criteria for PEW (body mass index, muscle mass loss, and deficient dietary intake), dedicated scoring systems have been developed to aid in assessing nutritional status. The two most important available scoring systems are the Subjective Global Assessment and the Kalantar score, also known as the Malnutrition Inflammation Score (MIS).

Subjective Global Assessment (SGA) is a tool that uses five components from the medical history (weight change, dietary intake, gastrointestinal symptoms, functional capacity, disease and their relation to nutritional requirements) and three components from a brief physical examination (signs of fat and muscle wasting, nutrition-associated alterations in fluid balance) to assess nutritional status. SGA has been validated in a variety of patient populations, including CKD patients [10].

The Malnutrition Inflammation Score (MIS) is also a validated nutritional assessment tool for CKD patients. MIS components incorporate anthropometric measurements, biochemical data including albumin, serum total iron binding capacity (TIBC), and components of the SGA. Each component is evaluated with four severity levels ranging from 0 (normal) to 3 (very severe). The total score is the sum of these 10 components ranging between 0 to 30. Both scoring systems have been developed to identify CKD patients with PEW/malnutrition [11].

Moreover, the mechanism of inflammation/malnutrition in CKD/ESKD is multifactorial. Proinflammatory cytokines induce anorexia, with associated increases in the incidence of chronic fatigue and the breakdown of muscle proteins, resulting in muscle atrophy. This anorexic process may be mediated by circulating appetite regulators, such as gastric mediators (e.g., leptin and ghrelin), adipokines, or cytokines [e.g., tumor necrosis factor (TNF), interleukin-6 (IL-6), or interleukin-1 (IL-1)] [2,5]. However, further research on these mediators is needed. In addition, other symptoms, and comorbidities, such as depression induced by IL-6, may lead to reduced nutrient intake, increased resting energy expenditure, suppression of anabolic hormones [such as growth hormone, insulin-like growth factor (IGF)-1, and testosterone] [2,12], or even resistance to hormonal factors [13,14,15,16,17].

Inflammasome-related biomarkers [IL-1, IL-1 receptor antagonist, IL-6, TNF-α, C-reactive protein (CRP), and fibrinogen] have been found to be inversely associated with the glomerular filtration rate (GFR) and positively associated with albuminuria [12]. Among these, IL-6 has been reported to be the best predictor of all-cause and cardiovascular mortality, with superior performance even to CRP and other cytokines, such as TNF-α, IL-1, and IL-18 [18,19]. In another study in a cohort of patients undergoing chronic hemodialysis (HD), CRP was found to be a better predictor of mortality than albumin or ferritin. Herselman et al. [3] demonstrated that CRP levels are not directly related to cardiovascular mortality, but a significant direct relationship was found with all-cause mortality. Finally, the role of adipokines in CKD has recently been acknowledged; for example, leptin has a potential inflammatory effect and is recognized as being involved in CKD progression and the promotion of oxidative stress, inflammation, and lipid disorders [20,21]. In CKD stages G2–5, increased leptin levels are associated with metabolic syndrome scores and higher plasma CRP [2,3,4,5], and they have also been described in HD patients with cachexia [20,21,22].

Other circulating biomarkers have been described in malnourished CKD patients. For instance, vitamin D has been shown to have anti-inflammatory effects (as well as magnesium and zinc), showing a lower activation as CKD progresses. Other components of chronic kidney disease-mineral and bone disorder, such as alkaline phosphatase, phosphate, and/or ionized calcium, have been associated with nutritional parameters. Similarly, omega-3 polyunsaturated fatty acids, known to have a strong anti-inflammatory effect, have lower levels in renal replacement therapy patients than in the general population [23].

Even though other sources of inflammation beyond CKD/uremic toxins (more than 25% of which are dietary o gut-derived) [24,25,26], such as infections or thrombotic events related to vascular access in HD patients, peritonitis in patients undergoing peritoneal dialysis, infections and immunosuppression in transplant patients, the intrinsic mechanisms associated with the development of a proinflammatory milieu is comparable in all CKD patients [2]. Further inflammatory factors are summarized in Figure 1.

The aforementioned factors together promote an inflammatory state that has a direct impact on the nutritional status of the patient, with repercussions for the quality of life and long-term prognosis. There is a need to improve nutrition in ESRD patients. Consequently, the use of oral meal supplementation, intradialytic and intraperitoneal parenteral nutrition may be efficaciously used to secure an adequate nutritional intake in order to prevent the breakdown of lean muscle mass, PEW, and cachexia [2].

### 2.1. Biomarkers of Nutritional Status in Haemodialysis Patients

Several studies have shown that patients who develop ESKD and undergo HD are prone to a progressive decline in nutritional parameters. The high mortality in HD patients is related, at least in part, to malnutrition and inflammation [2,5,12,27].

Nutritional assessment performed using MIS was significantly correlated with creatinine, C-reactive protein levels, and hematocrit. Furthermore, MIS showed a strong correlation with mortality in hemodialysis patients. Recently, the MIS score was also shown to be the best independent predictor of 4-year mortality in HD patients. In fact, MIS mortality predictability is equal to or exceeds many other complex tests of inflammation and nutritional status [28].

Among nutritional parameters, hypoalbuminemia has been the most extensively studied marker in dialysis patients, mainly because of its strong association with increased all-cause and cardiovascular mortality [1,2,5,27,29,30]. Although the discriminatory capacity of serum albumin (S-albumin) in identifying malnutrition is limited, it is of note that optimization of the S-albumin level nevertheless contributes to a reduction in mortality during the first year of HD [31]. The strong association between hypoalbuminemia and mortality in dialysis patients is well recognized, and it is at least partly explained by inflammation [3,31]. Not only is there a reduced amount of S-albumin in this population, but in addition, inflammation is associated with a decrease in albumin concentration as well as in albumin synthesis and half-life [12]. It is also to be noted that prealbumin, also known as transthyretin, the albumin precursor molecule, represents a commonly used marker for nutritional assessment in CKD, not only in HD patients. Prealbumin (transthyretin) is considered a sensitive marker for protein–energy status, but it is not widely available and more expensive than S-albumin. However, despite serum prealbumin providing prognostic value independent of S-albumin and other established predictors of mortality in HD patients (including acute kidney injury) [32,33], prealbumin levels have not been identified as an independent risk factor with respect to increased all-cause mortality [3,34]. In fact, the 2011 American Society for Parenteral and Enteral Nutrition’s (ASPEN’s) Clinical Guidelines advise that albumin and prealbumin not be used in isolation to assess nutrition status because they are markers of inflammatory metabolism [35]. Nevertheless, a correlation with several other biochemical markers has been described [36], and a reduction in transthyretin levels shows a linear correlation with other nutritional markers and systemic inflammation parameters, including energy and protein intake, as well as body fat and muscle mass [34,37]. Consequently, the recent 2020 Kidney Disease Outcomes Quality Initiative (KDOQI) guidelines for nutrition in CKD consider that albumin and/or serum prealbumin (if available), among other single biomarkers, may be considered *complementary* tools to assess nutritional status. However, it is stated that they should not be interpreted in isolation to assess nutritional status as they are influenced by non-nutritional factors [38,39,40].

It is less known that widely available and cheap serum creatinine (its association with nutrition is often disregarded) is a surrogate marker of skeletal muscle mass and dietary protein intake. Lower serum creatinine is clearly associated with an increased risk of death, and the creatinine level is also affected by systemic inflammation [41]. In the HD population, creatinine levels depend on residual renal function, dialysis dose, and even endogenous degradation [36].

Regarding the potential benefits of decreasing inflammatory molecules, a meta-analysis showed a reduction in inflammatory markers among adults with different stages of CKD who were submitted to a diet of non-animal origin [42]. Among the biomarkers assessed, CRP levels were higher in patients with animal protein consumption than in patients with isolated protein consumption. The limitation of this marker is its high variability, and it may overestimate or underestimate protein intake under catabolic or anabolic conditions, respectively [42]. On the other hand, interleukin-6 levels were found to be significantly increased in patients who consumed a diet rich in animal proteins compared to patients who consumed a vegetable-protein diet, probably related to the degree of renal insufficiency and the inflammation inherent to CKD, in addition to the type of protein consumed and catabolized [42]. Multiple studies have shown that high circulating levels of IL-6 contribute to inflammatory muscle protein losses [43] and may alter IGF-1 signaling by increasing the level of glucocorticoids. In uremic skeletal muscle, IL-6 has also been linked to increased caspase-3 activity (an initial step resulting in muscle protein loss) [44].

Special mention should be made of the TNF-related weak inducer of apoptosis (TWEAK), a member of the TNF superfamily that acts on responsive cells via binding to its receptor Fn14. TWEAK is linked to the signaling pathways involved in the regulation of nuclear factor kappa-light-chain enhancer of activated B cells (NF-κB), myogenesis, and apoptotic cascades [45]. In animal models, Fn14 induced tissue injury and inflammation, and a significant interaction was observed between soluble TWEAK and IL-6 in the prediction of mortality and reduced muscle strength in HD patients [46].

Lastly, leptin is a protein synthesized by adipocytes, skeletal muscle, and bone cells that regulates appetite and satiety, among other functions. Leptin has also been identified as a proinflammatory cytokine that shows a correlation with inflammation in obese patients. Leptin and its receptor share structural and functional properties with members of the cytokine family, and there is an inverse relationship between leptin levels and spontaneous dietary energy intake. A number of recent studies have shown that most CKD patients have inappropriately high leptin levels, and it has been speculated that leptin may be one of the factors mediating anorexia and wasting in this patient group. However, in addition to regulating appetite, leptin may also play a role in sympathetic activation, vascular calcification, oxidative stress, insulin secretion and sensitivity, bone formation, renal sodium handling, and hematopoiesis in CKD patients [47].

Taking all of the reported findings together, there are many data indicating a direct link between chronic inflammation and the development of malnutrition in ESKD patients. However, it is worth mentioning that the analysis of a single biomarker is probably insufficient for a correct assessment of nutritional status in HD patients and, in addition to normalized protein catabolic rate, combinations of different markers, such as S-albumin and/or prealbumin, CRP, creatinine and potentially others, may offer an improved prediction.

### 2.2. Biomarkers of Nutritional Status in Peritoneal Dialysis Patients

As is the case in the general population, peritoneal dialysis (PD) patients can be classified according to their nutritional pattern as anorexigenic/PEW, obese, or normal [48]. Depending on the pattern, they will have elevated or decreased inflammatory biomarkers. A study comparing these indicators showed that patients with PEW had higher TNF and IL-1 levels, while obese patients had the highest IL-6 level, and patients classified as “*normal*” had the highest adiponectin value [48]. CRP was elevated in all groups without any significant differences between them.

A well-known peculiarity of patients on PD, compared to patients on HD, is that their albumin values are lower, probably due to losses from peritoneal dialysis fluid (PDF) as well as the chronic inflammation status [48,49,50]. This finding is even more pronounced in diabetic patients [51].

The prevalence of PEW in PD patients was investigated in a meta-analysis and found to range from 16 to 98% (25th and 75th percentiles, 32 and 49%; median 36%) [52]. Moreover, patients on PD with PEW were found to have an inverse correlation between albumin and both CPR and IL-6 levels [49]. Interestingly, haptoglobin has also been described to be an inflammatory biomarker of particular value in non-diabetic patients on PD [53]. In addition, a negative correlation between albumin and inflammation burden has been demonstrated in this patient population [53].

In PD patients, a prospective cohort study confirmed that MIS is an independent predictor of fatal and nonfatal cardiovascular and infection events. Furthermore, this study showed that MIS is significantly correlated with early hospital admission and prolonged hospital stay. In another prospective study with 50 PD patients who were followed up for 6 months, it was shown that MIS was independently associated with a higher risk for future hospitalizations [54,55].

Up to 20% of patients starting PD gain more than 10% in body weight due to the energy in the form of glucose that is absorbed from the PDF [56]. It has been reported that a chronic inflammatory environment in obese patients on PD is mediated by CRP, fibrinogen, and leptin [56,57,58,59,60].

Other factors involved in PEW in PD patients are related to the technique itself. The exposure of PD patients to glucose as an osmotic agent may lead to the absorption of up to 300 g of glucose per day, depending on the patient’s membrane profile and the prescription of hypertonic solutions. Such a glucose load has a direct impact on the patient’s appetite, reducing the daily intake of proteins and other nutrients. In addition, there is a daily loss of protein through the peritoneal membrane, which in some cases may reach 10 g, and this, too, can contribute to a deterioration in nutritional status [61].

The nutritional situation of PD patients is, therefore, complex, as is the identification of inflammatory markers. Further and specific research is also needed in this area, focusing especially on preventive measures, given the negative impact of poor nutritional status on patient survival.

### 2.3. Biomarkers of Nutritional Status in Transplant Patients

The patients with CKD who undergo renal transplantation experience an important change in their nutritional balance. Metabolic stress may occur due to multiple factors: hormonal changes, alterations in nutrient intake and energy expenditure, loss of anorectic factors, low physical activity, immunosuppressive treatment, immune response to transplantation, episodes of rejection, deterioration in renal function, and the activation of systemic inflammation [52,53,54,55,56,57,58,59,60,61,62].

When there is a prior medical history of PEW in transplanted CKD patients, PEW may persist in 20–52% of patients, while others experience “malnutrition” characterized by obesity and metabolic syndrome [63]. Therefore, when evaluating the inflammatory factors related to nutritional status in the post-transplant period, patients should be classified into at least two groups: PEW and metabolic syndrome. In addition, in recent years, the measurement of muscle strength and mass has led to a re-emergence of the sarcopenia and dynapenia concepts. Classically, patients with sarcopenia had PEW, but a subset of patients displays a coexistence of obesity and muscular derangements, referred to as “sarcopenic obesity” and/or “dynapenic obesity”, which increases the diagnostic challenge [64]. Reduced muscle mass is a predominant phenotypic characteristic in stable renal transplant patients [64], but “dynapenic” and sarcopenic obesity have also been described in PD patients [58,65].

Few data are available on the impact of nutritional status on systemic inflammation in renal allograft recipients, and the methodology of the reported studies differs, hampering comparisons [66].

Molnar et al. evaluated the relationship between inflammation and PEW in a cohort of long-term renal transplant recipients in 2010, validating, for the first time, the low Malnutrition-Inflammation Score (MIS) in a cohort of renal transplant patients. In this study, it was observed that patients with a worse nutritional status had a worse renal function, were more often diabetic, and more frequently experienced graft rejection. As regards laboratory parameters, patients with a higher MIS score had lower hemoglobin, albumin, and prealbumin levels but higher transferrin, ferritin, IL-6, leptin, and TNF-α levels. CRP, serum IL-6, and TNF-α correlated positively with MIS grade, while abdominal circumference and prealbumin level correlated negatively [62].

Regarding obesity/metabolic syndrome, studies have shown that the prevalence of obesity triples after renal transplantation, from 6.3 to 18.8%, the increase is especially pronounced in women. An inflammatory pattern with significantly higher levels of IL-6 and CRP was described in obese patients versus patients with normal weight (defined as body mass index < 25 kg/m^2^). Moreover, CRP was related to the waist-to-hip ratio. These results suggest that fat, specifically visceral fat, is related to inflammation [66].

Leptin deserves special mention. The evolution of leptin levels during the first post-transplant year is variable: the level can increase, decrease, or remain stable. These changes are related to changes in body composition (visceral fat, fat mass) and to the patient’s phenotype [57,62]. Lower leptin values have been demonstrated following transplantation in patients with a pre-existing history of PEW and others with pretransplant obesity [57,62]. Several studies have correlated body mass index, body fat, and CPR. Although leptin has also been measured, study designs included the measurement of other factors in different months after transplantation. Consequently, the association between leptin and nutritional status after kidney transplantation needs further clarification [57].

As a novelty, some recent studies have correlated haptoglobin levels with metabolic syndrome. Previous studies have shown haptoglobin production by adipocytes from white adipose tissue, induced by proinflammatory cytokines such as IL-6 and TNF-α. These studies may indicate that haptoglobin could be a proinflammatory factor in situations without intravascular hemolysis, independently of other acute phase reactants such as CRP, procalcitonin, or albumin [67].

## 3. Conclusions

In summary, from the development of CKD and its progression, especially through advanced stages, a clear bidirectional association between inflammation and nutrition has been recognized. Multiple mechanisms seem to be involved, and therefore, the assessment and interpretation of overlapping nutritional and inflammatory parameters are complex. New evidence adapted to different patient statuses and kidney replacement therapies (HD, PD, kidney transplantation) is definitely needed to cast light on this interesting and important topic. Preventing malnutrition but also targeting inflammation seems to be important to improve the prognosis of CKD patients.

## Figures and Tables

**Figure 1 nutrients-14-04297-f001:**
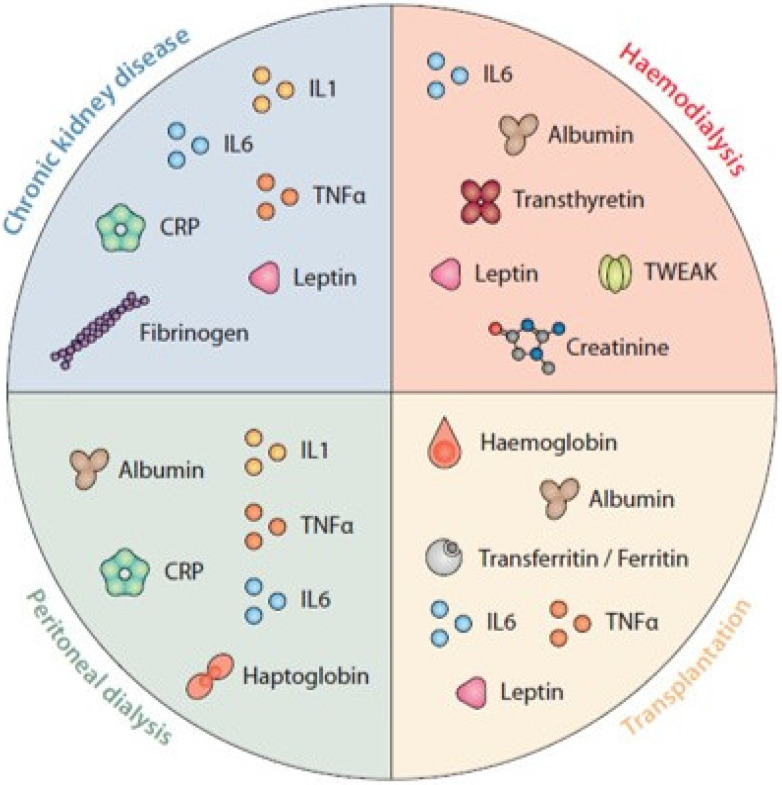
Biomarkers of nutritional status and inflammation. was designed and produced by Antonio Garcia from Bio-Graphics.

## Data Availability

Not applicable.

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
