# Peer review of "Evolving Concepts on Inflammatory Biomarkers and Malnutrition in Chronic Kidney Disease"

_nutrients, 2022, doi:10.3390/nu14204297_

Round 1

Reviewer 1 Report

Summary

Torres and colleagues have composed this review paper with the purpose of exploring currently available nutritional-inflammatory biomarkers in distinct CKD populations (haemodialysis, peritoneal dialysis, transplantation), with an objective to evaluating their efficacy as predictors of malnutrition and delineating their involvement in the common proinflammatory process. The authors further explain that while a direct relationship between inflammatory-nutritional status, signs and symptoms, and comorbidities has been established, there is still a need for clearly standardized markers of nutritional-inflammatory status in order to improve patient assessment and ultimately design appropriate treatment interventions. 

Overall this is a well-written review utilizing appropriate references. It covers a subject matter that is likely to be of interest in the scientific community broadly, and to nephrologists and nephrology researchers in particular, as well as endocrinologists, cardiologists, etc. Conversely, it is a subject that appears to have been covered not unfrequently in the last 7 years, in both primary research studies and potentially parallel reviews. Thus I am not entirely convinced that the paper as-written bridges a substantial knowledge gap in a manner sufficient to warrant publication in Nutrients. In addition, several seemingly important subtopics appear to be absent or incompletely addressed, in particular the MIS.   

General comments
The abstract provides a clear and succinct summary of the (presumptive) objectives and purpose of the review (ie; identifying and reviewing nutritional-inflammatory biomarkers likely to be pathologically relevant to CKD patients, with an intent to focus on their efficacy as predictors of malnutrition in this population specifically). In contrast the article Title and Introductory paragraphs tend to lose sight of the stated objectives, and thus the main purpose and justification for the article remains somewhat unclear until well into the body of the article. Suggest establishing earlier in the Introduction that there exists a complex interplay between overlapping nutritional and inflammatory parameters during CKD, and there is a need for further assessment and interpretation of these parameters as they relate to identifying and preventing malnutrition in particular.

The authors do a nice job of comparing and contrasting the biomarkers of interest in three distinct patient populations, however the near total lack of discussion related to Malnutrition Inflammation Score (MIS) seems a glaring omission. It is this reviewer’s current understanding that MIS has been shown to be a robust and clinically relevant marker of nutritional and inflammatory status in CKD populations. Importantly, its mortality predictability is equal to or exceeds many “elaborate tests of inflammation and nutritional status”, including quite a few of those covered in the present review. Thus the addition to the manuscript of a thorough review of MIS as it relates to the three patient sub-populations is strongly encouraged.

Less crucially, additional circulating markers may warrant mention, such as serum vitamin D levels, omega 3 FAs, dyslipidemia, and perhaps ionized calcium, alkaline phosphatase, anemia, Hb, ferritin, IL-10.

Finally, the relationships and predictive value of some MI biomarkers may differ substantially between pediatric and adult CKD patient populations. It may be of value to either expressly discuss established differences to date, or clearly specify that the current review relates to adult patient populations only.

Specific comments

Title: “Nutritional Aspects” is somewhat vague. Suggest clarifying further, for example “Nutritional Status” or perhaps “Inflammatory Biomarkers as Predictors of Malnutrition in…”?

CKD should be spelled out in full for the title (and all letters in the acronym should be capitalized.)

“Mechanisms of Inflammation…” (2nd section)

Second to last paragraph- I could not locate Figure 1. Has it been included somewhere with the submission?

Last paragraph- the final sentence is somewhat vague (“renal nutrition”) and lacking in sufficient foundation (“definitely needed to attenuate… ”)

“….in Haemodialysis …” (3rd section)

Second to last paragraph, last sentence- “leptin may play a role in…” Can you elaborate on what kind of role leptin may play specifically?

“Conclusions” (final section)

The final paragraph of the paper should be placed under a new subheading entitled “Conclusions” or “Summary” or similar.

Author Response

RESPONSE TO REVIEWER 1

Summary

Torres and colleagues have composed this review paper with the purpose of exploring currently available nutritional inflammatory biomarkers indistinct CKD populations (haemodialysis, peritoneal dialysis, transplantation), with an objective to evaluating their efficacy as predictors of malnutrition and delineating their involvement in the common proinflammatory process. The authors further explain that while a direct relationship between inflammatory-nutritional status, signs and symptoms, and comorbidities has been established, there is still a need for clearly standardized markers of nutritional-inflammatory status in order to improve patient assessment and ultimately design appropriate treatment interventions. 

Overall this is a well-written review utilizing appropriate references. It covers a subject matter that is likely to be of interest in the scientific community broadly, and to nephrologists and nephrology researchers in particular, as well as endocrinologists, cardiologists, etc. Conversely, it is a subject that appears to have been covered not unfrequently in the last 7 years, in both primary research studies and potentially parallel reviews. Thus I am not entirely convinced that the paper as-written bridges a substantial knowledge gap in a manner sufficient to warrant publication in Nutrients. In addition, several seemingly important subtopics appear to be absent or incompletely addressed, in particular the MIS.   

We thank the reviewer for recognizing that this is a well-written review utilizing appropriate references, and that it is likely to be of interest in the scientific community broadly, including specialties beyond Nephrology. On the other hand, we believe that we do contribute to “new” knowledge by underlining the (many times) unnoticed lack of homogeneity among different chronic kidney disease subpopulations (including renal replacement therapies), and the need of standardization.

We also want to thank the reviewer for the excellent contribution that we should expand and underline the important concept of malnutrition-inflammation syndrome (MIS). In fact, we completely agree that many pathophysiological pathways may share common pathophysiological routes (such as inflammation and/or MIS).

Therefore, as suggested, we have included several important changes. We now state that “Besides the diagnostic criteria for PEW (body mass index, muscle mass loss, and deficient dietary intake), dedicated scoring systems have been developed to aid in assessing nutritional status. The 2 most important available scoring systems are the Subjective Global Assessment and the Kalantar score also known as the Malnutrition Inflammation Score (MIS).

Subjective Global Assessment (SGA) is a tool that uses 5 components from the medical history (weight change, dietary intake, gastrointestinal symptoms, functional capacity, disease and their relation to nutritional requirements) and 3 components from a brief physical examination (signs of fat and muscle wasting, nutrition-associated alterations in fluid balance) to assess nutritional status. SGA has been validated in a variety of patient populations, including CKD patients [10].

Malnutrition Inflammation Score (MIS) is also a validated nutritional assessment tool for CKD patients. MIS components incorporate anthropometric measurements, biochemical data including albumin, serum total iron binding capacity (TIBC) and components of the SGA. Each component is evaluated with four severity levels ranging from 0 (normal) to 3 (very severe). Total score is the sum of these 10 components ranging between 0 to 30. Both scoring systems have been developed to identify CKD patients with PEW/malnutrition [11].” (Page 5, paragraph 1, line 90).

We additionally added as an important piece of information that “Nutritional assessment performed using MIS was significantly correlated with creatinine, C-reactive protein levels and hematocrit. Also, MIS showed a strong correlation with mortality in hemodialysis patients. Recently, MIS score was also shown to be the best independent predictor of 4-year mortality in HD patients. In fact, MIS mortality predictability is equal to or exceeds many other complex tests of inflammation and nutritional status”. (Page 9, paragraph 2, line 171) .

We also added that “In PD patients, a prospective cohort study confirmed that MIS is an independent predictor of fatal and nonfatal cardiovascular and infection events. Also, this study showed that MIS is significantly correlated with early hospital admission and prolonged hospital stay.  In another prospective study with 50 PD patients who were followed up for 6 months it was shown that MIS was independently associated with a higher risk for future hospitalizations” (Page 13, paragraph 4, line 283).

General comments
The abstract provides a clear and succinct summary of the (presumptive) objectives and purpose of the review (ie; identifying and reviewing nutritional-inflammatory biomarkers likely to be pathologically relevant to CKD patients, with an intent to focus on their efficacy as predictors of malnutrition in this population specifically). In contrast the article Title and Introductory paragraphs tend to lose sight of the stated objectives, and thus the main purpose and justification for the article remains somewhat unclear until well into the body of the article. Suggest establishing earlier in the Introduction that there exists a complex interplay between overlapping nutritional and inflammatory parameters during CKD, and there is a need for further assessment and interpretation of these parameters as they relate to identifying and preventing malnutrition in particular.

We thank the reviewer for considering necessary this introductory link

We added “In fact, there is a complex interplay between overlapping nutritional and inflammatory parameters during CKD, and there is a need for further assessment and interpretation of these parameters as they relate to identifying and preventing malnutrition in particular” (Page 4, paragraph 1, line 70).

The authors do a nice job of comparing and contrasting the biomarkers of interest in three distinct patient populations, however the near total lack of discussion related to Malnutrition Inflammation Score (MIS) seems a glaring omission. It is this reviewer’s current understanding that MIS has been shown to be a robust and clinically relevant marker of nutritional and inflammatory status in CKD populations. Importantly, its mortality predictability is equal to or exceeds many “elaborate tests of inflammation and nutritional status”, including quite a few of those covered in the present review. Thus the addition to the manuscript of a thorough review of MIS as it relates to the three patient sub-populations is strongly encouraged.

We again thank the reviewer for this important comment. Notice in the previous response that we added, “MIS mortality predictability is equal to or exceeds many other complex tests of inflammation and nutritional status”. (Page 9, paragraph 2, line 171) .

Less crucially, additional circulating markers may warrant mention, such as serum vitamin D levels, omega 3 FAs, dyslipidemia, and perhaps ionized calcium, alkaline phosphatase, anemia, Hb, ferritin, IL-10.

We agree with the reviewer that at least vitamin D (and potentially other biomarkers of the chronic kidney disease-mineral and bone disorders) as well as omega-3 polyinsaturated fatty acids may warrant mention. Therefore, we now state that:

 “Other circulating biomarkers have been described in CKD malnourished patients. For instance, vitamin D has been shown to have anti-inflammatory effects (as well as magnesium and zinc), showing a lower activation as CKD progresses. Other components of the chronic kidney disease-mineral and bone disorder, such as alkaline phosphatase, phosphate and/or ionized calcium have been associated with nutritional parameters. or Similarly, omega-3 polyunsaturated fatty acids, known to have a strong anti-inflammatory effect, have lower levels in renal replacement therapy patients than in the general population” (Page 7, paragraph 2, line 140) .

Finally, the relationships and predictive value of some MI biomarkers may differ substantially between pediatric and adult CKD patient populations. It may be of value to either expressly discuss established differences to date, or clearly specify that the current review relates to adult patient populations only.

This is a very well taken point. As suggested, we now state, “This review aims to explore the currently available nutritional-inflammatory biomarkers in adults with a view to evaluating their efficacy as predictors of malnutrition and their involvement in the proinflammatory process in this patient population” (Page 4, paragraph 1, line 73)

Specific comments

Title: “Nutritional Aspects” is somewhat vague. Suggest clarifying further, for example “Nutritional Status” or perhaps “Inflammatory Biomarkers as Predictors of Malnutrition in…”?

As per a nice suggestion of the reviewer, we changed the previous title for “Evolving concepts on inflammatory biomarkers and malnutrition in chronic kidney disease”.

CKD should be spelled out in full for the title (and all letters in the acronym should be capitalized.)

Corrected

“Mechanisms of Inflammation…” (2nd section)

Second to last paragraph- I could not locate Figure 1. Has it been included somewhere with the submission?

Sent. It is likely that we had a problem during the previous uploading process and we apologize for that.

Last paragraph- the final sentence is somewhat vague (“renal nutrition”) and lacking in sufficient foundation (“definitely needed to attenuate… ”)

As suggested by the Reviewer we changed the sentence to “There is a need to improve nutrition in ESRD patients. Consequently, the use of oral meal supplementation, intradialytic and intraperitoneal parenteral nutrition may be efficaciously used to secure an adequate nutritional intake in order to prevent breakdown of lean muscle mass, PEW and cachexia” (Page 8, paragraph 2, line 162)

“….in Haemodialysis …” (3rd section)

Second to last paragraph, last sentence- “leptin may play a role in…” Can you elaborate on what kind of role leptin may play specifically?

As suggested by the Reviewer we specified the missing information. We now state,”A number of recent studies have shown that most CKD patients have inappropriately high leptin levels, and it has been speculated that leptin may be one factor mediating anorexia and wasting in this patient group. However, besides regulating appetite, leptin may also play a role in sympathetic activation, vascular calcification, oxidative stress, insulin secretion and sensitivity, bone formation, renal sodium handling, and hematopoiesis in CKD patients” (Page 12, paragraph 2, line 245)

“Conclusions” (final section)

The final paragraph of the paper should be placed under a new subheading entitled “Conclusions” or “Summary” or similar.

Corrected. As suggested by the Reviewer, we included a subheading (Page 17, paragraph 3, line 369).

Reviewer 2 Report

This review by Torres et al. discusses the various inflammatory markers associated with nutritional status in different CKD/ESRD stages and dialysis modality.

Major comments:

The review has several syntax errors that could be improved by someone fluent in English.

The review is not conceptually novel and does not add to what is already known about the complex interplay between the uremic milieu, inflammation, and nutrition.  Rather than just a summary of the various literature regarding inflammatory biomarkers and a call for standardized markers to assess nutritional inflammation, the authors should editorialize better what specific research can be performed to answer these questions. 

For example, in the section regarding CRP levels being higher in patients with animal protein consumption than in patients with isolated protein consumption, has there been any study illustrating switching these patients to a non-animal protein diet with similar caloric value changes the CRP levels?  If not, then the authors should raise this as a needed study.

Minor comment:

Figure 1 is missing - I was not able to find it in the manuscript document or elsewhere.

Author Response

RESPONSE TO REVIEWER 2

This review by Torres et al. discusses the various inflammatory markers associated with nutritional status in different CKD/ESRD stages and dialysis modality.

Major comments:

The review is not conceptually novel and does not add to what is already known about the complex interplay between the uremic milieu, inflammation, and nutrition.  Rather than just a summary of the various literature regarding inflammatory biomarkers and a call for standardized markers to assess nutritional inflammation, the authors should editorialize better what specific research can be performed to answer these questions. For example, in the section regarding CRP levels being higher in patients with animal protein consumption than in patients with isolated protein consumption, has there been any study illustrating switching these patients to a non-animal protein diet with similar caloric value changes the CRP levels?  If not, then the authors should raise this as a needed study.

We thank the reviewer for this excellent suggestion. We do believe that we contribute to “new” knowledge by underlining the many times unnoticed lack of homogeneity among different chronic kidney disease subpopulations (including renal replacement therapies). In accordance with reviewer 1, we therefore agree that it is necessary to expand the concept of malnutrition-inflammation syndrome (MIS), and we adapted the paper accordingly.  Moreover, the specific suggested research (analyzing inflammation parameters such as CRP when switching patients to a non-animal protein diet) represents an excellent add-on to the review. It may be now particularly relevant when several nephrology guidelines start advocating for changing old paradigms about plant-derived food in nephrology patients.

We added the following paragraphs:

“Besides the diagnostic criteria for PEW (body mass index, muscle mass loss, and deficient dietary intake), dedicated scoring systems have been developed to aid in assessing nutritional status. The 2 most important available scoring systems are the Subjective Global Assessment and the Kalantar score also known as the Malnutrition Inflammation Score (MIS).

Subjective Global Assessment (SGA) is a tool that uses 5 components from the medical history (weight change, dietary intake, gastrointestinal symptoms, functional capacity, disease and their relation to nutritional requirements) and 3 components from a brief physical examination (signs of fat and muscle wasting, nutrition-associated alterations in fluid balance) to assess nutritional status. SGA has been validated in a variety of patient populations, including CKD patients [10].

Malnutrition Inflammation Score (MIS) is also a validated nutritional assessment tool for CKD patients. MIS components incorporate anthropometric measurements, biochemical data including albumin, serum total iron binding capacity (TIBC) and components of the SGA. Each component is evaluated with four severity levels ranging from 0 (normal) to 3 (very severe). Total score is the sum of these 10 components ranging between 0 to 30. Both scoring systems have been developed to identify CKD patients with PEW/malnutrition [11].” (Page 5, paragraph 1, line 90).

We additionally added as an important piece of information that “Nutritional assessment performed using MIS was significantly correlated with creatinine, C-reactive protein levels and hematocrit. Also, MIS showed a strong correlation with mortality in hemodialysis patients. Recently, MIS score was also shown to be the best independent predictor of 4-year mortality in HD patients. In fact, MIS mortality predictability is equal to or exceeds many other complex tests of inflammation and nutritional status”. (Page 9, paragraph 2, line 171) .

We also added thatIn PD patients, a prospective cohort study confirmed that MIS is an independent predictor of fatal and nonfatal cardiovascular and infection events. Also, this study showed that MIS is significantly correlated with early hospital admission and prolonged hospital stay.  In another prospective study with 50 PD patients who were followed up for 6 months it was shown that MIS was independently associated with a higher risk for future hospitalizations” (Page 13, paragraph 4, line 283)

“There is a need to improve nutrition in ESRD patients. Consequently, the use of oral meal supplementation, intradialytic and intraperitoneal parenteral nutrition may be efficaciously used to secure an adequate nutritional intake in order to prevent breakdown of lean muscle mass, PEW and cachexia” (Page 8, paragraph 2, line 162)

Minor comment:

Figure 1 is missing - I was not able to find it in the manuscript document or elsewhere.

Sent. It is likely that we had a problem during the previous uploading process and we apologize for that.

Round 2

Reviewer 1 Report

All of my suggestions have been adequately addressed by the authors.

One minor comment regarding Figure 1: the yellow quadrant heading (I believe it's "Transplantation") is quite difficult to see. Suggest shadowing the font or using a darker colour.